# Association between Breast Cancer Polygenic Risk Score and Chemotherapy-Induced Febrile Neutropenia: Null Results

**DOI:** 10.3390/cancers14112714

**Published:** 2022-05-31

**Authors:** Seeu Si Ong, Peh Joo Ho, Alexis Jiaying Khng, Elaine Hsuen Lim, Fuh Yong Wong, Benita Kiat-Tee Tan, Swee Ho Lim, Ern Yu Tan, Su-Ming Tan, Veronique Kiak Mien Tan, Rebecca Dent, Tira Jing Ying Tan, Joanne Ngeow, Preetha Madhukumar, Julie Liana Bte Hamzah, Yirong Sim, Geok Hoon Lim, Jinnie Siyan Pang, Veronica Siton Alcantara, Patrick Mun Yew Chan, Juliana Jia Chuan Chen, Sherwin Kuah, Jaime Chin Mui Seah, Shaik Ahmad Buhari, Siau Wei Tang, Celene Wei Qi Ng, Jingmei Li, Mikael Hartman

**Affiliations:** 1Women’s Health and Genetics, Genome Institute of Singapore, 60 Biopolis Street, Genome, #02-01, Singapore 138672, Singapore; ong_seeu_si@gis.a-star.edu.sg (S.S.O.); hopj@gis.a-star.edu.sg (P.J.H.); khngja@gis.a-star.edu.sg (A.J.K.); 2Department of Surgery, Yong Loo Lin School of Medicine, National University of Singapore, Singapore 119228, Singapore; ephbamh@nus.edu.sg; 3Saw Swee Hock School of Public Health, National University of Singapore, Singapore 117549, Singapore; 4Division of Medical Oncology, National Cancer Centre Singapore, Singapore 169610, Singapore; elaine.lim.hsuen@singhealth.com.sg (E.H.L.); rebecca.dent@duke-nus.edu.sg (R.D.); tira.tan.j.y@singhealth.com.sg (T.J.Y.T.); joanne.ngeow@ntu.edu.sg (J.N.); 5Division of Radiation Oncology, National Cancer Centre Singapore, Singapore 169610, Singapore; wong.fuh.yong@singhealth.com.sg; 6Division of Surgery and Surgical Oncology, National Cancer Centre Singapore, Singapore 169610, Singapore; benita.tan.k.t@singhealth.com.sg (B.K.-T.T.); veronique.tan.k.m@singhealth.com.sg (V.K.M.T.); madhukumar.preetha@singhealth.com.sg (P.M.); julie.liana.hamzah@singhealth.com.sg (J.L.B.H.); sim.yirong@singhealth.com.sg (Y.S.); 7Department of Breast Surgery, Singapore General Hospital, Singapore 169608, Singapore; 8Department of General Surgery, Sengkang General Hospital, Singapore 544886, Singapore; 9KK Breast Department, KK Women’s and Children’s Hospital, Singapore 229899, Singapore; lim.swee.ho@singhealth.com.sg (S.H.L.); lim.gh@singhealth.com.sg (G.H.L.); jinnie.pang.sy@singhealth.com.sg (J.S.P.); veronica.siton.alcantara@kkh.com.sg (V.S.A.); 10Department of General Surgery, Tan Tock Seng Hospital, Singapore 308433, Singapore; ern_yu_tan@ttsh.com.sg (E.Y.T.); patrick_chan@ttsh.com.sg (P.M.Y.C.); juliana_jc_chen@ttsh.com.sg (J.J.C.C.); sherwinkuah@gmail.com (S.K.); 11Lee Kong Chian School of Medicine, Nanyang Technological University, Singapore 308232, Singapore; 12Institute of Molecular and Cell Biology, Singapore 138673, Singapore; 13Division of Breast Surgery, Changi General Hospital, Singapore 529889, Singapore; tan.su.ming@singhealth.com.sg (S.-M.T.); jaime_seah@cgh.com.sg (J.C.M.S.); 14Department of Surgery, University Surgical Cluster, National University Health System, Singapore 119228, Singapore; sursasb@nus.edu.sg (S.A.B.); siau_wei_tang@nuhs.edu.sg (S.W.T.); celene_wq_ng@nuhs.edu.sg (C.W.Q.N.)

**Keywords:** febrile neutropenia, neutropenic fever, breast cancer, polygenic risk score, PRS

## Abstract

**Simple Summary:**

The most common complication of chemotherapy for cancer patients is febrile neutropenia (FN). This is an abnormally low blood neutrophil count coupled with a fever that leaves patients susceptible to fatal infections. Genetic variants for breast cancer risk linked to chemotherapy-induced toxicity have been previously explored. We study the association between a validated 313 genetic marker-based breast cancer polygenic risk score (PRS) and chemotherapy-induced neutropenia without fever, and febrile neutropenia (FNc) in Asian breast cancer patients treated with chemotherapy. PRS distributions were not significantly different in any of the comparisons. Higher PRS_overall_ quartiles were 9% less likely to develop neutropenia, and 13% less likely to develop FNc. However, the associations were not statistically significant. No dose-dependent trend was observed for the estrogen receptor- (ER-) positive weighted PRS (PRS_ER-pos_) and ER-negative weighted PRS (PRS_ER-neg_). Breast cancer PRS was not strongly associated with chemotherapy-induced neutropenia or FNc.

**Abstract:**

Background: The hypothesis that breast cancer (BC) susceptibility variants are linked to chemotherapy-induced toxicity has been previously explored. Here, we investigated the association between a validated 313-marker-based BC polygenic risk score (PRS) and chemotherapy-induced neutropenia without fever and febrile neutropenia (FNc) in Asian BC patients. Methods: This observational case-control study of Asian BC patients treated with chemotherapy included 161 FNc patients, 219 neutropenia patients, and 936 patients who did not develop neutropenia. A continuous PRS was calculated by summing weighted risk alleles associated with overall, estrogen receptor- (ER-) positive, and ER-negative BC risk. PRS distributions neutropenia or FNc cases were compared to controls who did not develop neutropenia using two-sample *t*-tests. Odds ratios (OR) and corresponding 95% confidence intervals were estimated for the associations between PRS (quartiles and per standard deviation (SD) increase) and neutropenia-related outcomes compared to controls. Results: PRS distributions were not significantly different in any of the comparisons. Higher PRS_overall_ quartiles were negatively correlated with neutropenia or FNc. However, the associations were not statistically significant (PRS per SD increase OR neutropenia: 0.91 [0.79–1.06]; FNc: 0.87 [0.73–1.03]). No dose-dependent trend was observed for the ER-positive weighted PRS (PRS_ER-pos_) and ER-negative weighted PRS (PRS_ER-neg_). Conclusion: BC PRS was not strongly associated with chemotherapy-induced neutropenia or FNc.

## 1. Introduction

Febrile neutropenia (FN) occurs when a patient develops a fever when the count of neutrophils in the blood is abnormally low (neutropenia) [1]. The resulting impaired ability to respond to inflammation puts the patient at an increased risk of a life-threatening infection.

FN is one of the most common complications of myelosuppressive cancer chemotherapy, often with clinically significant treatment delay and dose reduction as consequences [2,3]. As treatment efficacy may be compromised, FN poses a major problem to patient morbidity and survival [4]. Some reports in the literature note a higher FN incidence in Asian patients compared to Caucasian patients [5,6]. The FN-related mortality rate of breast cancer patients has been reported from 2.6% to 5.6% but is higher among older patients [7,8]. The incidence of FN is reported to be 10–20% in breast cancer patients treated with adjuvant anthracycline-based chemotherapy [9,10,11]. Without prophylactic treatment using granulocyte colony-stimulating factor (G-CSF), up to one in four breast cancer patients treated with taxanes or anthracyclines can develop FN [12]. In particular, taxane and anthracycline-based chemotherapy regimens were found to be clinically predictive of developing FN in a retrospective study of cancer patients in Singapore, of which more than 60% were breast cancer patients [13].

Chemotherapy drugs have a narrow therapeutic index, with a small difference between the dose required for cancer treatment and which leads to adverse side effects [14]. The ability to stratify each patient according to their individual risk of developing FN is thus clinically useful. Certain patient characteristics are known to have a higher risk of FN than others. Risk factors that have been reported include older age, late-stage disease, comorbidities, low baseline cytopenia, and low body surface area/body mass index (BMI) [15]. A low BMI (<23 kg/m^2^), in particular, was shown to increase FN risk by over fourfold in an Asian breast cancer cohort [11].

The elevated production of reactive oxygen species as a result of an inflammatory response can result in oxidative DNA damage [16]. As such, genetic variants in cancer pathways, such as DNA repair, may be able to modify the cytotoxic effects associated with chemotherapy [17,18]. For example, certain common polymorphisms in DNA repair genes can impair the removal of DNA adducts, which may adversely affect the response to chemotherapy and result in neutropenia [19].

To date, large genome-wide association studies have identified a large number of single nucleotide polymorphism (SNP) markers that are predictive of breast cancer risk [20,21]. The effects of multiple individual SNPs, which are typically small, can be summarised into a single polygenic risk score (PRS). A 313-SNP PRS for breast cancer developed using 94,075 breast cancer cases and 75,017 controls from 69 studies has been validated in large populations to be a reliable and robust tool for estimating an individual’s breast cancer risk due to common genetic variants [22]. In this study, we examine the relationship between breast cancer PRS and the risk of developing FN after chemotherapy treatment in Asian breast cancer patients.

## 2. Materials and Methods

### 2.1. Study Population

The study population was derived from the patients enrolled in the Singapore Breast Cancer Cohort (SGBCC) who were diagnosed with or treated for breast cancer (ICD9: 174; ICD 10: C50) in five restructured hospitals (National University Hospital (NUH), KK Women’s and Children’s Hospital (KKH), Tan Tock Seng Hospital (TTSH), Singapore General Hospital (SGH), or National Cancer Centre Singapore (NCCS)). Patient information on lifestyle and demographic variables were collected from questionnaires. Tumour characteristics and treatment details were retrieved from hospital medical records. All studies were performed in accordance with the Declaration of Helsinki and all participants provided written informed consent. SGBCC was approved by the National Healthcare Group Domain Specific Review Board (reference number: 2017/00797) and the SingHealth Centralised Institutional Review Board (reference number: 2016/3010).

The combination of tumour histologic grade, and immunohistochemical markers for estrogen receptor (ER), progesterone receptor (PR), and human epidermal growth factor receptor 2 (HER2), were used to define the breast cancer tumour proxy subtypes: luminal A (ER+/PR+, HER2−, and well- or moderately differentiated); luminal B (HER2 negative) (ER+/PR+, HER2−, and poorly differentiated); luminal B (HER2 positive) (ER+/PR+, HER2+, and poorly differentiated); HER2-overexpressed (HER2+); triple negative (ER−, PR−, and HER2−) [23].

### 2.2. Analytica Dataset

Adult patients (n = 1596) in SGBCC who were diagnosed between 2000 and 2016 with invasive breast cancer and who received adjuvant or neoadjuvant chemotherapy regimens containing taxane or anthracycline were included in this study. Patients (n = 441) who received G-CSF at any time during treatment or within 30 days prior to the first chemotherapy date were excluded (Appendix A).

### 2.3. Outcome of Interest

Hospitalisations for FN (neutropenia (ICD 9: 288, ICD 10: D70) with fever (ICD 9: 780, ICD 10: R50.9)) were retrospectively identified through the hospital discharge summaries (KKH, SGH, and NCCS) or manually extracted from medical records (NUH and TTSH). The main outcome of interest was the occurrence of FN from initiation of chemotherapy treatment (using taxanes or anthracyclines) to 30 days from the last chemotherapy treatment cycle (i.e., within 30 days of last chemotherapy treatment) (FNc). Neutropenia with or without fever was examined as an alternative outcome. Breast cancer patients who developed FNc or any neutropenia were compared to breast cancer patients who did not develop any neutropenia.

### 2.4. Genotyping and Imputation of Common Variants

Genomic DNA was extracted using standard methods from whole blood and saliva samples. Briefly, FlexiGene DNA Kit (Qiagen, Germantown, MD, USA, Catalogue number 51206) was used for genomic DNA extraction from buffy coats isolated from whole blood samples according to the manufacturer’s protocol; prepIT-L2P DNA extraction kit (DNA Genotek, Kanata, ON, Canada, Catalogue number PT-L2P-45) was used for genomic DNA extraction from saliva samples collected with Oragene^®^ DNA saliva collection kit (DNA Genotek, Canada, Catalogue number OG-500) according to the manufacturer’s protocol. The manufacturer’s protocol for genomic DNA extraction is described in further detail in Appendix A. The Illumina Infinium OncoArray 500 K BeadChip was used for genotyping DNA samples [24]. Detailed information on genotype calling and quality control has been described previously [20]. Imputation was performed in two parts: SHAPEIT (v2.r904) was used for phasing [25,26] and IMPUTE2 (v2.3.2) was used for the imputation [27]. The October 2014 (version 3) release of the 1000 Genomes Project was the reference panel used [28]. All chromosomal positions described are in reference to the human genome assembly GRCh37 (hg19).

### 2.5. Polygenic Risk Score (PRS)

Calculation of the breast cancer PRS was performed using a weighted summation of risk alleles as described in Mavaddat, N, et al. [22]. A list of the included SNPs and their corresponding effect sizes and weights are shown in Appendix A. Overall weights were used to calculate the overall PRS (PRS_overall_); estrogen receptor (ER)-positive weights for the ER-positive PRS (PRS_ER-pos_); ER-negative weights for the ER-negative PRS (PRS_ER-neg_).

### 2.6. Statistical Analysis

Demographics, lifestyle, and tumour characteristics were compared between breast cancer patients who developed FNc and those who did not develop neutropenia (the Kruskal–Wallis test and Chi-square test were used for continuous and categorical variables, respectively). Data fulfil the non-normal assumption for the Kruskal–Wallis test. Odds ratios and corresponding confidence intervals were estimated for each of the 313 breast cancer SNPs using the Plink software (v1.90b5.2, --fisher command). Logistic regression models were used to estimate odds ratios and corresponding confidence intervals for the association between breast cancer PRS and FNc. Residuals were inspected. Variables known from literature as risk factors for FN and those found to be significantly (*p* < 0.05) associated with FNc were included for adjustment in multivariable logistic models. Missing data were treated as a separate category. The analyses were repeated for cases defined as neutropenia with or without fever compared to non-neutropenic controls.

All analyses were performed with R (version 4.0.2) unless otherwise stated. Datasets utilised in analysis can be made available upon request.

## 3. Results

Breast cancer patients who did not receive prophylactic G-CSF (median age = 52 years) were significantly younger than those who received G-CSF ≤ 30 days before the start of chemotherapy (median age = 55 years) or those who received G-CSF after chemotherapy initiation (median age = 52 years) (Appendix A). Other variables found to be significantly different between the included and excluded subjects include recruitment site, year of diagnosis, ethnicity, and tumour stage and grade (Appendix A).

A total of 1155 breast cancer patients remained after the exclusion of all patients who received G-CSF (Table 1). A total of 219 patients were identified to have developed neutropenia based on ICD codes in their medical reports. Of these patients who developed neutropenia, 161 developed FNc. Patients did not develop neutropenia and were treated as controls in this study (n = 936).

The median age at breast cancer diagnosis was 52 years (interquartile range (IQR): 46 to 59) and was not significantly different between patients who developed FNc and those who did not (*p* = 0.393). Patients of Malay ethnicity were more likely to develop FNc as compared to Chinese patients (Chi-square test comparing the two ethnicity groups, *p* = 0.007).

### 3.1. Single SNP Analyses

Figure 1 and Appendix A show the results of the associations between 313 SNPs included in the breast cancer PRS and FNc and neutropenia, respectively. Of the 313 SNPs analysed, 15 (4.8%) were associated with FNc at *p* < 0.05. For FNc, the smallest *p*-value observed (*p* = 0.00193) was for rs7842619_G_T (base-pair position 124739913, Appendix A No. 162) annotated to *ANXA13*. A total of 14 SNPs (4.5%) were associated with neutropenia with or without fever. For any neutropenia, the smallest *p*-value observed (*p* = 0.00669) was for rs13147907_T_A (base-pair position 187503758, Appendix A No. 78), an intergenic variant. None of the observed associations were significant after correcting for multiple comparisons.

### 3.2. Association between PRS and FNc

The distribution of PRS (overall, ER-positive, ER-negative) was not significantly different between patients who developed FNc and patients who did not (Table 1 and Figure 2). Although there was a dose-dependent protective trend, per standard deviation (SD) increase in PRS_overall_ was not significantly associated with FNc (adjusted OR: 0.84 [0.71–1.00], *p* = 0.055) (Table 2). No clear dose-dependent effect was shown across quartiles for the associations between PRS_ER-pos_ and PRS_ER-neg_. However, the effect sizes observed for per SD increases in PRS_ER-pos_ (adjusted OR: 0.84 [0.71 to 1.01], *p* = 0.057) and PRS_ER-neg_ (adjusted OR: 0.86 [0.72 to 1.02], *p* = 0.080) were found to be similar to that for PRS_overall_ (Table 2). Further adjustment for tumour characteristics significantly associated with neutropenia-related outcomes (Table 1) did not appreciably change the results (Appendix A).

### 3.3. Associations between PRS and Neutropenia (with or without Fever)

The association between PRS_overall_ and neutropenia (with or without fever) was attenuated when compared to FNc (Table 2). Adjusted OR per SD increases associated with any neutropenia was 0.90 [0.77 to 1.05] (*p* = 0.173). No clear dose-dependent trend was observed for the associations between breast cancer subtype-weighted PRS (PRS_ER-pos_ and PRS_ER-neg_) and any neutropenia-related outcome.

## 4. Discussion

Due to the detrimental impact that FN has on patient management and treatment, risk assessment of patients predisposed to serious complications from well-established breast cancer chemotherapy treatment regimens is of importance in decreasing morbidity and mortality. Predictive models built using clinical parameters, such as the Multinational Association for Supportive Care in Cancer (MASCC) risk index score, have been shown to be widely applicable in stratifying patients by their individual FN risk [29]. In validation studies for MASCC where a majority of the subjects did not have haematological malignancies, sensitivity is above 80%, while specificity ranges from 36 to 64% [29]. Furthermore, the MASCC risk index score together with patient-reported outcome measures for neutropenia was found to perform at a higher specificity with fewer misclassifications for determining FN risk amongst cancer patients in Singapore [30]. A separate study in Singapore of cancer patients, whereby more than 60% of the study population were breast cancer patients, found liver and renal function tests to be potential predictors of FN risk, requiring further in-depth evaluation of their prognostic value [13].

Assessment of risk of developing FN determines prophylactic G-CSF strategy. Previous studies have indicated primary prophylactic G-CSF for elderly breast cancer patients to be effective in reducing FN incidence (24% to 6%), reducing the number of patients with clinically significant dose delays (21% to 14%) or dose reductions (32% to 15%), and even increasing the number chemotherapy cycles administered (by 2%) [31,32]. Collectively, these effects can potentially improve chemotherapy treatment efficacy and disease outcomes for breast cancer patients. However, comorbidities (hypertension, hyperlipidaemia, cardiovascular disease and diabetes mellitus) remain risk factors for developing FN, in spite of prophylactic G-CSF [3,13]. Additionally, the controversial use of adjunctive G-CSF for FN treatment did significantly reduce neutropenia-related mortality but did not significantly reduce hospitalisation length of stay or duration for neutrophil count recovery [33].

Genetics is an integral part of personalised therapeutics. Previous studies have explored the involvement of candidate genes in the aetiology of FN. In a study of 216 breast cancer patients treated with anthracycline-based chemotherapy, Okishiro et al. reported significant associations between *MDM2* SNP309 and *TP53* R72P with severe neutropenia and FN, respectively [34]. *MDM2* and *TP53* are genes known to play a part in resistance to chemotherapy [34]. In another study comprising 100 breast cancer patients (18% developed FN), Awada and colleagues looked for FN markers using a genotyping chip with customised content on drug metabolising enzymes and transporters [35]. Genetic variants in seven genes (*ABCC6*, *ABCG1*, *ABCG2*, *CYP1A2*, *CYP2D6*, *FMO2*, and *FMO3*) were found to be significantly associated.

A shared genetic component in the aetiologies between breast cancer and FN is possible through common pathways such as DNA repair [36]. Some studies have reported that breast cancer patients who are germline carriers of pathogenic variants in DNA repair genes such as *BRCA1* and *BRCA2* are at higher risk of FN [36]. However, the finding is not always consistent. Others have found no appreciable difference in FN incidence between carriers and non-carriers [37].

The search for FN genetic markers has been extended to include other breast cancer predisposition markers. In a previous work looking at common breast cancer genetic variants and chemotherapy toxicities, Dorling et al. examined the association between 94 SNPs and their aggregate PRS in 499 cases (neutropenia grades 3 and 4) and 1177 controls (neutropenia grades 0–2) [38]. A high breast cancer genetic risk (PRS weighted by the per-allele log-odds ratio for risk of breast cancer associated with each of the 94 variants) was found to be protective against chemotherapy-induced neutropenia, but the result was not significant.

In spite of the inclusion of more SNPs and a different reference group, our results are in the same direction as that reported by Dorling et al. in that breast cancer PRS may be protective against neutropenia [38]. Nonetheless, PRS is disease-specific. The odds that breast cancer and chemotherapy-induced febrile neutropenia would share a genetic risk score are low. In addition, the breast cancer PRS includes variants that are not only involved in DNA repair. Variants in genes which modify breast cancer risk may also not necessarily modify treatment response or trigger adverse effects [39]. More relevant gene variants include polymorphic variants in absorption, distribution, metabolism, and excretion (ADME) genes for which the taxanes or anthracyclines are the substrates [40]. Other relevant polymorphic variants include genes involved in immune system activity since the patients were treated with immunosuppressive regimens [41]. Currently, the evidence remains limited as to whether breast cancer patients with a high genetic predisposition for the disease are associated with a different risk of developing FNc complications from chemotherapy treatment than other patients with lower genetic risk.

There are several limitations to our study. Patients who received chemotherapy form a very heterogeneous group. Due to the retrospective nature of this study, details on neutrophil counts prior to chemotherapy, treatment dosage, drug combinations, blood pressure, and comorbidities were not available. Nonetheless, dose density is dependent on body mass index, which we controlled for in the analyses. Tumour characteristics significantly associated with the development of FNc were also adjusted for in the analyses. This study included only patients hospitalised for neutropenia, hence some cases may be missed. The MASCC risk index (FN risk assessment using clinical factors) for individual patients cannot be computed. Some characteristics of patients who were excluded due to the receipt of G-CSF were different from the analytical population, which may result in selection bias. However, the result will be biased to the null. Nonetheless, our results cannot be generalised to patients who receive prophylactic G-CSF as they are identified as high risk prior to the start of chemotherapy. The adverse events studied are limited to neutropenia with or without fever. Future analyses may consider an expanded list of other adverse effects commonly associated with chemotherapy treatment in breast cancer patients.

## 5. Conclusions

The 313-SNP breast cancer polygenic risk score was not found to be strongly associated with FNc or neutropenia in this study of 1596 breast cancer patients of Asian descent in Singapore.

## Figures and Tables

**Figure 1 cancers-14-02714-f001:**
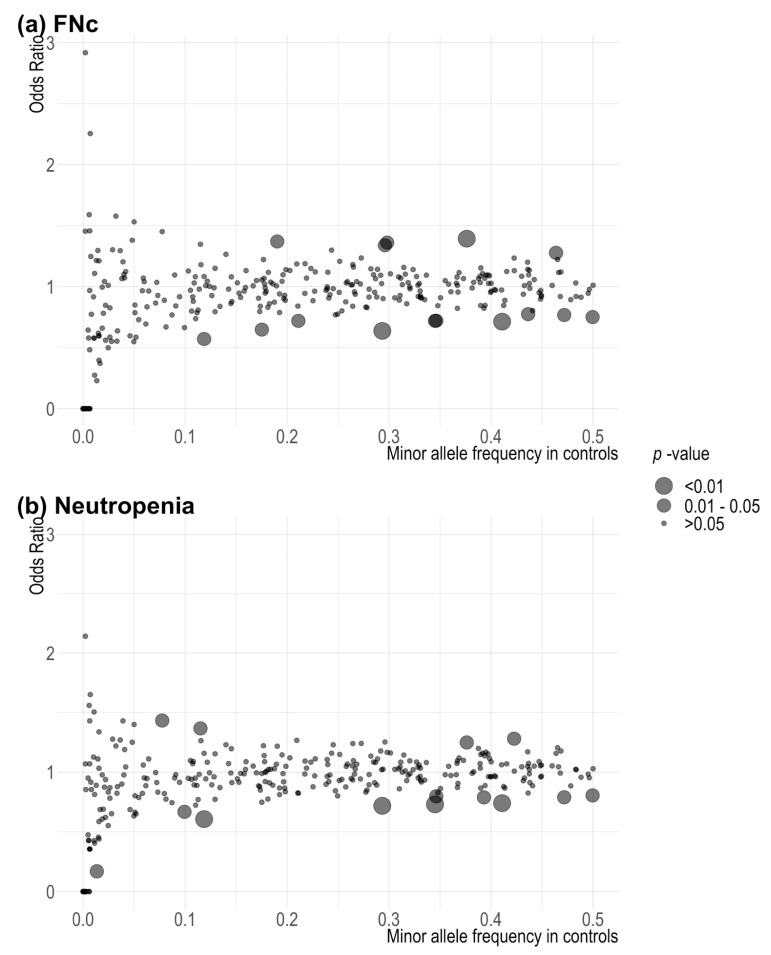
Scatter plot of *p*-values associated with 313 single variants in the breast cancer polygenic risk score in (**a**) FNc: febrile neutropenia from initiation of chemotherapy treatment (using taxanes or anthracyclines) to 30 days from last chemotherapy treatment cycle (i.e., within 30 days of last chemotherapy treatment); (**b**) neutropenia. Fisher’s exact test was used to estimate the associations.

**Figure 2 cancers-14-02714-f002:**
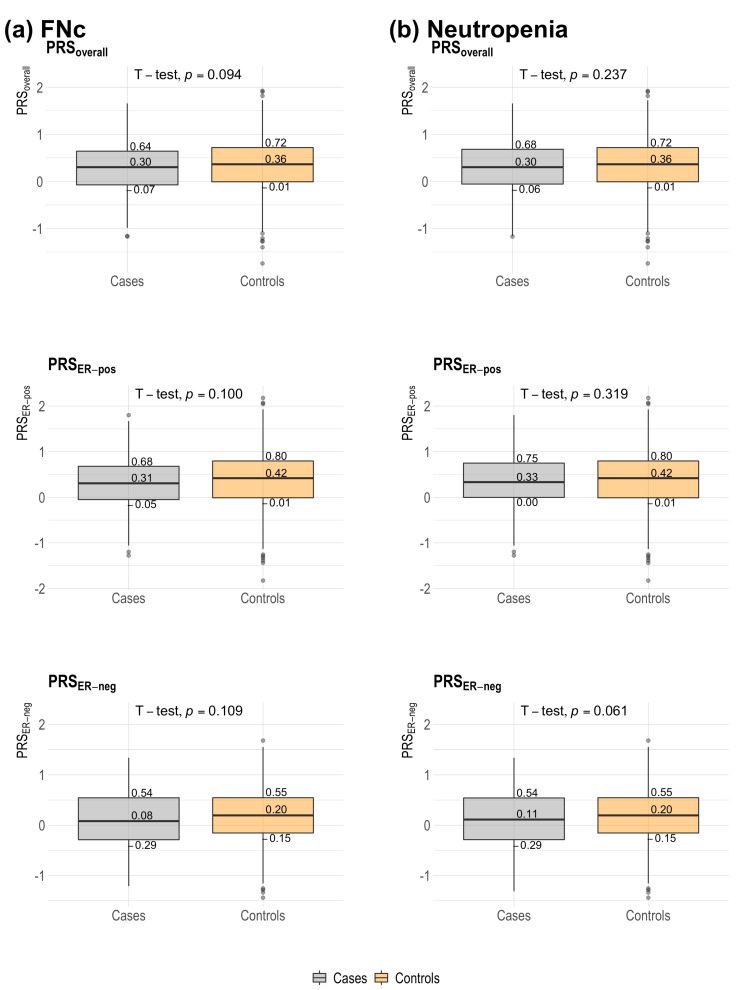
Distribution of 313-SNP breast cancer polygenic risk score (overall, estrogen receptor (ER)-positive, ER-negative) among breast cancer patients with neutropenia-related outcomes and non-neutropenia controls. Controls were chemotherapy-treated breast cancer patients who did not develop neutropenia. Median scores and quartile scores are displayed. A two-sample *t*-test was used for comparisons of medians; *p*-values are displayed. (**a**) FNc: febrile neutropenia from initiation of chemotherapy treatment (using taxanes or anthracyclines) to 30 days from last chemotherapy treatment cycle (i.e., within 30 days of last chemotherapy treatment); (**b**) neutropenia. PRS: polygenic risk score; PRS_overall_: overall PRS; ER: estrogen receptor; PRS_ER-pos_: ER-positive PRS; PRS_ER-neg_: ER-negative PRS.

**Table 1 cancers-14-02714-t001:** Description of the analytical cohort of breast cancer patients treated with chemotherapy. ^1^ Comparison with controls who did not develop neutropenia using Kruskal–Wallis test for continuous variables, and Chi-square test for categorical variables. FNc: febrile neutropenia from initiation of chemotherapy treatment (using taxanes or anthracyclines) to 30 days from last chemotherapy treatment cycle (i.e., within 30 days of last chemotherapy treatment); IQR: interquartile range; KKH: KK Women’s and Children’s Hospital; NUH: National University Hospital; SGH: Singapore General Hospital; NCCS: National Cancer Centre Singapore; TTSH: Tan Tock Seng Hospital; HER2: human epidermal growth factor receptor 2; SNP: single nucleotide polymorphism; ER: estrogen receptor.

	Total n = 1155	No Neutropenia n = 936	FNc n = 161	*p* ^1^	Neutropenia n = 219	*p* ^1^
**Demographics**						
**Median age at diagnosis (years, IQR)**	52 (46–59)	52 (45–59)	52 (47–58)	0.381	52 (47–59)	0.391
**Case type (n, %)**						
Incident	592 (51)	487 (52)	76 (47)	0.296	105 (48)	0.311
Prevalent	563 (49)	449 (48)	85 (53)		114 (52)	
**Recruitment Site (n, %)**						
KKH	98 (8)	94 (10)	1 (1)	<0.001	4 (2)	<0.001
NUH	643 (56)	496 (53)	119 (74)		147 (67)	
SGH and NCCS	261 (23)	223 (24)	14 (9)		38 (17)	
TTSH	153 (13)	123 (13)	27 (17)		30 (14)	
**Year of diagnosis (n, %)**						
Before 2005	89 (8)	68 (7)	18 (11)	0.175	21 (10)	0.442
2005–2010	400 (35)	329 (35)	59 (37)		71 (32)	
2011–2016	666 (58)	539 (58)	84 (52)		127 (58)	
**Ethnicity (n, %)**						
Chinese	871 (75)	717 (77)	113 (70)	<0.001	154 (70)	<0.001
Malay	227 (20)	166 (18)	46 (29)		61 (28)	
Indian	57 (5)	53 (6)	2 (1)		4 (2)	
**Body mass index in kg/m^2^ (n, %)**						
<20	124 (11)	96 (10)	21 (13)	0.275	28 (13)	0.240
20–24	540 (47)	450 (48)	63 (39)		90 (41)	
25–29	337 (29)	266 (28)	51 (32)		71 (32)	
>30	141 (12)	117 (12)	21 (13)		24 (11)	
Unknown	13 (1)	7 (1)	5 (3)		6 (3)	
**Tumour characteristics**						
**Tumour stage (n, %)**						
I	207 (18)	173 (18)	26 (16)	0.117	34 (16)	0.038
II	539 (47)	452 (48)	63 (39)		87 (40)	
III	278 (24)	211 (23)	48 (30)		67 (31)	
IV	77 (7)	63 (7)	10 (6)		14 (6)	
Unknown	54 (5)	37 (4)	14 (9)		17 (8)	
**TNM tumour size (n, %)**						
≤20 mm	379 (33)	316 (34)	47 (29)	0.003	63 (29)	<0.001
21–50 mm	497 (43)	414 (44)	60 (37)		83 (38)	
>50 mm	114 (10)	88 (9)	21 (13)		26 (12)	
Attached to chest wall	76 (7)	49 (5)	19 (12)		27 (12)	
Unknown	89 (8)	69 (7)	14 (9)		20 (9)	
**Nodal status (n, %)**						
Positive	474 (41)	390 (42)	62 (39)	0.769	84 (38)	0.725
Negative	592 (51)	481 (51)	82 (51)		111 (51)	
Unknown	89 (8)	65 (7)	17 (11)		24 (11)	
**Grade (n, %)**						
Well-differentiated	78 (7)	67 (7)	9 (6)	0.066	11 (5)	0.006
Moderately differentiated	397 (34)	338 (36)	47 (29)		59 (27)	
Poorly differentiated	595 (52)	461 (49)	97 (60)		134 (61)	
Unknown	85 (7)	70 (7)	8 (5)		15 (7)	
**Estrogen receptor status (n, %)**						
Positive	759 (66)	631 (67)	93 (58)	0.005	128 (58)	0.005
Negative	355 (31)	269 (29)	66 (41)		86 (39)	
Unknown	41 (4)	36 (4)	2 (1)		5 (2)	
**Progesterone receptor status (n, %)**						
Positive	696 (60)	579 (62)	87 (54)	0.024	117 (53)	0.010
Negative	416 (36)	319 (34)	72 (45)		97 (44)	
Unknown	43 (4)	38 (4)	2 (1)		5 (2)	
**HER2 status (n, %)**						
Positive	341 (30)	277 (30)	47 (29)	0.943	64 (29)	0.950
Negative	642 (56)	519 (55)	91 (57)		123 (56)	
Unknown	172 (15)	140 (15)	23 (14)		32 (15)	
**Proxy subtype (n, %)**						
Luminal A	307 (27)	263 (28)	35 (22)	0.013	44 (20)	0.004
Luminal B (HER2–ve)	232 (20)	183 (20)	33 (20)		49 (22)	
Luminal B (HER2+ve)	135 (12)	114 (12)	16 (10)		21 (10)	
HER2-overexpressed	128 (11)	100 (11)	20 (12)		28 (13)	
Triple negative	138 (12)	98 (10)	32 (20)		40 (18)	
Missing	215 (19)	178 (19)	25 (16)		37 (17)	
**313-SNP breast cancer polygenic risk score (median (IQR))**						
Overall	0.351 (−0.017–0.715)	0.364 (−0.009–0.718)	0.302 (−0.074–0.642)	0.131	0.302 (−0.057–0.682)	0.275
ER-positive	0.399 (−0.010–0.788)	0.419 (−0.012–0.797)	0.306 (−0.051–0.680)	0.162	0.332 (−0.003–0.750)	0.389
ER-negative	0.184 (−0.179–0.545)	0.195 (−0.154–0.546)	0.082 (−0.288–0.545)	0.094	0.112 (−0.287–0.542)	0.075

**Table 2 cancers-14-02714-t002:** Association between 313-single nucleotide polymorphism (SNP) breast cancer polygenic risk score and neutropenia-related outcomes in chemotherapy-treated breast cancer patients who did not receive granulocyte colony-stimulating factor (G-CSF). Controls were chemotherapy-treated breast cancer patients who did not develop neutropenia. ^1^ Adjusted for recruitment site, ethnicity, body mass index and population structure (first four principal components). FNc: febrile neutropenia from initiation of chemotherapy treatment (using taxanes or anthracyclines) to 30 days from last chemotherapy treatment cycle (i.e., within 30 days of last chemotherapy treatment); OR: odds ratio; CI: confidence interval; SNP: single nucleotide polymorphism; PRS: polygenic risk score; PRS_overall_: overall PRS; ER: estrogen receptor; PRS_ER-pos_: ER-positive PRS; PRS_ER-neg_: ER-negative PRS.

	FNc n = 161				Neutropenia n = 219			
	Crude		Adjusted ^1^		Crude		Adjusted ^1^	
	OR (95% CI)	*p*	OR (95% CI)	*p*	OR (95% CI)	*p*	OR (95% CI)	*p*
313-SNP Breast Cancer Polygenic Risk Score								
**PRS_overall_**								
Quartile 1	1.00 (Reference)		1.00 (Reference)		1.00 (Reference)		1.00 (Reference)	
Quartile 2	0.92 (0.59–1.45)	0.723	0.88 (0.55–1.41)	0.588	1.04 (0.69–1.55)	0.863	0.99 (0.66–1.05)	0.968
Quartile 3	0.84 (0.54–1.33)	0.462	0.81 (0.51–1.30)	0.389	0.84 (0.56–1.27)	0.412	0.84 (0.55–1.29)	0.435
Quartile 4	0.64 (0.39–1.05)	0.079	0.57 (0.35–0.96)	0.033	0.78 (0.51–1.19)	0.243	0.73 (0.47–1.13)	0.154
Continuous, per SD increase	0.87 (0.73–1.03)	0.097	0.84 (0.71–1.00)	0.055	0.91 (0.79–1.06)	0.235	0.90 (0.77–1.05)	0.173
**PRS_ER-pos_**								
Quartile 1	1.00 (Reference)		1.00 (Reference)		1.00 (Reference)		1.00 (Reference)	
Quartile 2	1.14 (0.72–1.79)	0.583	1.12 (0.70–1.81)	0.631	1.24 (0.83–1.87)	0.291	1.25 (0.82–1.90)	0.301
Quartile 3	0.94 (0.59–1.50)	0.800	0.90 (0.55–1.46)	0.662	0.93 (0.61–1.42)	0.749	0.93 (0.60–1.44)	0.745
Quartile 4	0.77 (0.47–1.26)	0.305	0.71 (0.42–1.18)	0.184	0.93 (0.60–1.42)	0.722	0.88 (0.57–1.37)	0.578
Continuous, per SD increase	0.87 (0.74–1.03)	0.105	0.84 (0.71–1.01)	0.057	0.93 (0.80–1.08)	0.320	0.91 (0.78–1.06)	0.232
**PRS_ER-neg_**								
Quartile 1	1.00 (Reference)		1.00 (Reference)		1.00 (Reference)		1.00 (Reference)	
Quartile 2	0.58 (0.36–0.93)	0.025	0.57 (0.35–0.93)	0.024	0.61 (0.40–0.92)	0.019	0.61 (0.39–0.93)	0.023
Quartile 3	0.54 (0.34–0.87)	0.012	0.56 (0.34–0.92)	0.023	0.60 (0.40–0.91)	0.017	0.62 (0.40–0.95)	0.027
Quartile 4	0.71 (0.46–1.10)	0.129	0.66 (0.42–1.05)	0.077	0.72 (0.49–1.07)	0.102	0.71 (0.47–1.06)	0.092
Continuous, per SD increase	0.86 (0.73–1.02)	0.089	0.86 (0.72–1.02)	0.080	0.86 (0.74–1.00)	0.045	0.86 (0.73–1.00)	0.046

## Data Availability

The data underlying this article will be shared upon reasonable request to the study principal investigator (Email: ephbamh@nus.edu.sg).

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
