# Peer review of "Association between Breast Cancer Polygenic Risk Score and Chemotherapy-Induced Febrile Neutropenia: Null Results"

_cancers, 2022, doi:10.3390/cancers14112714_

Round 1
Reviewer 1 Report
The authors analyzed febrile neutropenia as one of adverse effects accompanying chemotherapy of breast cancer based on taxanes and anthracyclines. They performed the analysis based on meta-analysis published in the paper: Mavaddat N, Michailidou K, Dennis J et al. Polygenic Risk Scores for Prediction of Breast Cancer and Breast Cancer Subtypes. 406 American Journal of Human Genetics, 2019, 104(1), 21-34, and analyzed themselves 1596 samples from breast cancer patients. The authors submitted the list of polymorphic variants under analysis together with statistics. They also collected clinical data which, opposite to genetic analyses, showed significance in respect to the occurrence of febrile neutropenia and neutropenia. The authors took the same set of polymorphic variants which was used to predict the risk of developing breast cancer and molecular subtypes of breast cancer. The null result they got in their analysis can be connected with the list of polymorphic variants they chose for analysis. The genes which variants modify the cancer risk not necessarily modify the response to treatment or the presence of adverse effects. Instead they should choose the polymorphic variants in ADME genes for which the taxanes or anthracyclines are the substrates or polymorphic variants of genes involved in immune system activity since the patients were treated with immunosuppressive regimens.
I have doubts concerning the proper design of the study. In such analyses usually the whole set of adverse effects is taken to statistical analysis, fox example according to EGOG Common Toxicity Criteria.
Reviewer 2 Report
The manuscript titled “Association between breast cancer polygenic risk score and chemotherapy-induced febrile neutropenia: Null results” describes a 1596 breast cancer patients’ clinical investigation. A total of 313 single nucleotide polymorphisms were used to calculate the breast cancer polygenic risk score. The authors concluded there was no strong relation between 313-SNP breast cancer PRS and FCs/neutropenia in this study. Overall, the manuscript is well-written and prepared. The followings are some concerns and comments have been pointed out that the authors may want to consider before acceptance.
Concerns and Comments:
- Line 56: None “neutropenic fever” appears in the main context, I’d suggest the authors use another one due to it is not suitable to be a keyword. PRS is the abbreviation of polygenic risk score, I’d suggest the authors take them as one keyword; add some other keywords if the authors prefer.
- BC abbreviates to breast cancer has not been used in the main context. It’s OK. But it seems there is no need only define it in the abstract.
- Lines 61-64: I’d suggest the authors add references to the first sentence of this paragraph or the paragraph.
- Supplementary Table 1: I’d suggest the authors use “number (percentage)” in the second row to make the information clearer, for example, “Total n=1,596 patients (%)” or add “%” to all the numbers in the bracket of the table, for example, “1367 (86%)”. Or whatever format that the authors preferred. Please double-check and make necessary to Table 1 as well.
- Line 139: Provide detailed information on “FlexiGene DNA Kit”, for example, CAT#, etc. Please check throughout the manuscript.
- Lines 138-148: Please add a similar description, “the protocols were followed by the manufacturers’ instructions”. Or briefly describe protocols instead of listing each kit name.
- I’d suggest the authors use italic p as it refers to a p-value. Please check throughout the manuscript, for example, line 165, Supplementary Table 1, and so on.
- Line 173: Please use “years old” instead of only “years”. Check throughout the manuscript.
- Please consistent the format with or without space before and after the equal symbol throughout the manuscript. For example, Table 2, “n=161”, “n = 219”.
- Table 2: I’d suggest the authors just use “Ref.” in the table and define “Ref.: reference” at the bottom of the table, or adjust the size of each column to make it looks better instead of some cells in one line some cells contain two lines.
- Line 327: The 1,596 breast cancer patients in this study could not present all the breast cancer patients in Asia. The conclusion was too broad.
- Please release all the authors’ names in the references (besides those with a super big group of authors) instead of using “et al.”. For example, reference 4, reference 5, reference 10, reference 11, and so on. Check all the references.
Round 2
Reviewer 1 Report
The authors did not change anything in the manuscript. They added only one sentence from my review and the appropriate citation. I stand by my recommendation.
Reviewer 2 Report
Please make some necessary corrections and check throughout the manuscript.
- Line 169, line 204, line 230 Figure 2 in the images: The p here should be italic.
- Supplementary Table 3 and Table 4: In the first row of the Table, it should be “n=219”, but the equal signs are missing.
